# Radical Hysterectomy in Early-Stage Cervical Cancer: Abandoning the One-Fits-All Concept

**DOI:** 10.3390/jpm13091292

**Published:** 2023-08-24

**Authors:** Tommaso Bianchi, Tommaso Grassi, Luca Bazzurini, Giampaolo Di Martino, Serena Negri, Robert Fruscio, Gaetano Trezzi, Fabio Landoni

**Affiliations:** 1Department of Medicine and Surgery, University of Milano-Bicocca, 20126 Milano, Italy; t.bianchi4@campus.unimib.it (T.B.); s.negri20@campus.unimib.it (S.N.); robert.fruscio@unimib.it (R.F.); fabio.landoni@unimib.it (F.L.); 2Clinic of Obstetrics and Gynecology, IRCCS Fondazione San Gerardo dei Tintori, 20900 Monza, Italy; bazzuriniluca@hotmail.com (L.B.); giamp.dima@gmail.com (G.D.M.); gaetano.trezzi@gmail.com (G.T.)

**Keywords:** cervical cancer, radical hysterectomy, tailoring radicality, parametrectomy, surgical oncology, morbidity

## Abstract

Two pillars in modern oncology are treatment personalization and the reduction in treatment-related morbidity. For decades, the one-fits-all concept of radical hysterectomy has been the cornerstone of early-stage cervical cancer surgical treatment. However, no agreement exists about the prevalent method of parametrial invasion, and the literature is conflicting regarding the extent of parametrectomy needed to achieve adequate surgical radicality. Therefore, authors started investigating if less radical surgery was feasible and oncologically safe in these patients. Two historical randomized controlled trials (RCTs) compared classical radical hysterectomy (RH) to modified RH and simple hysterectomy. Less radical surgery showed a drastic reduction in morbidity without jeopardizing oncological outcomes. However, given the high frequency of adjuvant radiotherapy, the real impact of reduced radicality could not be estimated. Subsequently, several retrospective studies investigated the chance of tailoring parametrectomy according to the tumor’s characteristics. Parametrial involvement was shown to be negligible in early-stage low-risk cervical cancer. An observational prospective study and a phase II exploratory RCT have recently confirmed the feasibility and safety of simple hysterectomy in this subgroup of patients. The preliminary results of a large prospective RCT comparing simple vs. radical surgery for early-stage low-risk cervical cancer show strong probability of giving a final answer on this topic.

## 1. Introduction

Despite a drastic reduction in incidence and mortality, cervical cancer still represents a major cause of death worldwide, especially in developing countries [1]. Thanks to the diffusion of screening programs, most cases are diagnosed when the disease is confined to the cervix. Radical hysterectomy (RH) with sentinel lymph node (SLN) biopsy and bilateral pelvic lymphadenectomy represents the standard treatment for women with no child-bearing desire and early-stage cervical cancer [2]. However, the literature is conflicting with regard to the extent of the radicality of the surgery, and no conclusive data can be extrapolated from the two historical randomized controlled trials (RCTs) that compared less extensive with more extensive radical surgery. Therefore, no strong evidence exists to recommend the extent of the parametrectomy needed to achieve adequate surgical radicality.

## 2. Radical Parametrial Resection: Pros and Cons

### 2.1. Radical Hysterectomy Classification

The term “radical hysterectomy” defines the wide variety of procedures requiring the removal of the uterus together with the anterior, posterior, and lateral parametria. Since the parametrial tissue is the most frequent site of local diffusion of cervical carcinoma, the rationale is to resect the cervix with healthy tissue borders as far away as possible from the tumor. Depending on the presence of risk factors such as tumor size, lymphovascular space invasion (LVSI), depth of cervical invasion, and pelvic node metastases, clinically unidentified parametrial spreading may be identified at the final pathology report in up to 30% of patients with early-stage tumors (tumor size < 4 cm confined to the cervix) [3,4].

The extent of the parametrial resection affects the radicality of the procedure. Many modifications of the historical Wertheim’s RH technique [5] have been proposed in the last century to improve oncologic outcomes and decrease surgery-related morbidity [6,7,8,9,10,11]. Many different classifications have been introduced to standardize the technique and properly describe the extent of radicality. In 1974, the Piver–Rutledge classification [6] was used to describe the radicality of hysterectomy (Table 1). However, it lacked a rigorous adherence to international anatomical definitions or clear anatomical landmarks and was scarcely reproducible. Therefore, in 2008, Querleu and Morrow introduced their novel four-tier internationally accepted classification, which was revised in 2017 (Table 2) [7,8].

### 2.2. Type of Parametrial Invasion: How Cervical Cancer Invades the Parametrium

Parametrial invasion can occur (i) continuously, via direct infiltration of paracervical tissue, or (ii) discontinuously, via tumor emboli within parametrial lymphatic vessels and parametrial node involvement. However, no agreement exists on the prevalent way cervical cancer metastasizes in the parametrium nor as to its main pattern of parametrial spread (anterior, posterior, lateral diffusion of the disease). Thus, the different opinions about parametrial function reflect the debate on the ideal extent of radical surgery in early cervical cancer treatment.

Historically, complete removal of the parametria was recommended, according to the evidence that parametrial metastasis can occur equally in both the lateral and medial parametria. In 1978, Burghardt et al. introduced a technique to process the parametria using giant sections. They demonstrated the existence of lymph nodes with a scattered and unpredictable location within the parametrium, and positive parametrial lymph nodes were detected even in patients with small cervical tumors. Thus, they advocated complete parametrectomy as the cornerstone of surgical management of cervical carcinoma [12,13]. Similarly, Benedetti Panici et al. analyzed the surgical specimens of 69 patients who underwent Piver class III radical hysterectomy and bilateral pelvic lymphadenectomy for stage IB–IIA cervical carcinoma. The parametrial processing technique described by Burghardt et al. was used. They found parametrial lymph nodes scattered in all the parametrial ligaments examined in 93% of patients. The most frequent site of metastases was the lateral parametrium, mostly due to tumor emboli in lymphovascular spaces and parametrial lymph node metastases rather than contiguous invasion, independently of the tumor size [3].

Other authors expressed some concerns about the radical removal of the parametrium. In 1995, Sartori et al. [14] studied parametrial node involvement in 215 patients with clinical stage IB cervical cancer treated by primary Piver class III abdominal radical hysterectomy. The authors reported tumor size as a strong predictive factor of parametrial involvement, since no positive parametrial nodes were detected in patients with tumor size < 2 cm. Most importantly, they found positive parametrial nodes in 2% and 3% of patients in the distal (lateral) and proximal (medial) part of the lateral parametrium, respectively. Therefore, they suggested that a less radical surgical procedure (i.e., Piver class II radical hysterectomy) may cause an inadequate tumor resection in the lateral parametrium in only 2% of patients. However, a “geographic omission” in 2% of patients corresponded to a “therapeutic omission” in less than 1% of patients: in their series, 62.5% of cases with distal parametrial node involvement had pelvic lymph node metastases and, thus, required adjuvant treatment. Additionally, in other case series, the direct parametrial invasion was the most common way of parametrial invasion of cervical cancer, demonstrating little clinical relevance in removing the most lateral part of the parametrium [4,15,16,17]. Winter et al. [4] examined the surgical specimens of 351 node-negative patients with FIGO stage IB, IIA, or IIB cervical carcinoma who underwent Piver class III radical hysterectomy. Medial parametrium was the most commonly involved via direct infiltration or discontinuously via tumor emboli and parametrial node involvement. In contrast, isolated discontinuous involvement of the most lateral portion of the parametrium occurred in only six cases (1.7%). Similarly, Puente et al. [15] evaluated 107 patients with FIGO IB1-IIA cervical carcinoma. They found parametrial involvement in 14.9% of cases, mostly (62.5%) by direct extension in the internal (i.e., medial) parametrium. External parametrial involvement was found in 3.7% of patients (4/107).

### 2.3. Pattern of Parametrial Spread: Which Way the Tumor Prefers (Anterior, Posterior, Lateral Diffusion of the Disease)

Landoni et al. [18] evaluated the surgical specimens of 230 patients who underwent Piver class III RH for stage IB–IIA cervical cancer between 1989 and 1993. They aimed to define the prevalent direction of tumor spread within and outside the cervical stroma. Twelve giant sections were obtained from every surgical specimen, each including cervical and paracervical tissues (a “compass-rose” evaluation of the specimen). Tumor spread within the cervical stroma was found in all directions. In 28% and 27% of cases, the maximum depth of stromal invasion of the cervix was found in the anterior and posterior quadrants, respectively, while 22% was found in each lateral one. In patients with parametrial involvement, a positive anterior parametrium (vesico-cervical ligaments and vesico-cervical septum) was found in 23% of cases, while posterior, right, and left lateral parametria were involved in 15%, 28%, and 34% of cases, respectively. In their series, a minimum thickness of tumor-free cervical stroma (less than 3 mm) was a predictive factor of parametrial involvement. The authors suggested that this parameter could reflect the carcinomatous stromal invasion better than the absolute value of the maximum depth for two main reasons. First, tumor invasion is related to the thickness of the cervix, which may be influenced by age and parity. The ratio between tumor and cervical size is an essential predictor of the risk of parametrial spread, as already described by Burghardt et al. [9], who described tumor size in terms of tumor–cervix quotient. Since the tumor potential of invading paracervical tissues is related to cervical thickness, the minimum thickness of tumor-free cervical stroma best reflects the tumor’s proximity to the parametria rather than the depth of invasion. Second, the involvement of anterior and posterior paracervical tissues was almost invariably associated with a minimum thickness of unaffected stroma of less than 3 mm in the corresponding front and back quadrants. In particular, this occurred in 92% of cases with anterior tumor spread through the vesico-cervical ligaments and the vesico-cervical septum. The evidence on the prevalent tumor growth through the antero-posterior axis within the cervix and the corresponding extra-cervical extension into paracervical tissues that are not widely included by conventional RH started suggesting the limits of surgery. Therefore, wide resection of the lateral parametrium as in the conventional RH may be useless; if adequate surgical margins cannot be achieved anteriorly, adjuvant treatment is then mandatory.

### 2.4. Morbidity and Oncologic Safety of Modified Radical Hysterectomy

Parametrectomy is the main cause of intra- and post-operative complications in the surgical treatment of cervical carcinoma. The incidence of surgery-related morbidity is irrespective of the surgical approach [19]. RH may be associated with large volume blood loss, lower urinary tract, rectal, and sexual dysfunction. The reduction in blood supply, direct surgical trauma, subsequent perivisceral fibrosis, and interruption of the autonomic nerves contained in the parametrium are the most frequent causes of surgical morbidity [20,21,22,23]. The nerve branches run along the deepest/caudal portion of the lateral parametrium and the anterior parametrium’s most-lateral part. Attempts have been made to preserve the autonomic plexus to decrease the incidence of adverse effects by reducing the radicality of the procedure without jeopardizing oncologic outcomes. In the last three decades, the literature data provided evidence that modified RH (i.e., Piver class II/ Q-M type B) and nerve-sparing RH (i.e., Q-M type C1) are safe in terms of oncological outcomes, compared to classical RH (i.e., Piver class III/Q-M type C2) in the treatment of early-stage tumors. Additionally, they are even superior in terms of morbidity, therefore questioning the need for a more radical resection of the parametrium [24,25,26,27,28].

In the 1990s, evidence started arising regarding the therapeutic appropriateness and oncologic safety of modified RH. Like Photopulos’ previous findings [29], Magrina et al. retrospectively evaluated the oncological and morbidity outcomes in patients with IA–IB2 cervical carcinoma treated with modified RH and bilateral lymphadenectomy. In their series, they registered shorter operative time and postoperative hospital stay compared to their historical class III RH cohort. Additionally, they described a very low (<1%) risk of significant urinary tract complications and no pelvic recurrences among the patients with tumors 4 cm or less in size. They claimed the feasibility and oncological safety of the central resection provided by modified RH in that specific group of patients [24,30]. In line with these data, Landoni et al. published the sole prospective randomized study comparing Piver class II to Piver class III RH [25]. Two-hundred and thirty-eight patients with stage IB1–IB2 cervical carcinoma were randomized. No differences in pelvic recurrence rate (24% in class II vs. 26% in class III), disease-free survival (75% class II vs. 73% class III), and overall survival (81% in class II vs. 77% class III) were observed. Indeed, the amount and degree of morbidities following RH were strictly related to the extent of resection. They observed a significantly lower mean operative time and a lower rate of vesical disorders in the group undergoing class II RH. However, some limitations of this study must be considered [31]. First, 55% of patients received adjuvant radiotherapy in both arms—up to 80% if we consider patients with tumors > 4 cm in size only. The high frequency of adjuvant radiotherapy may interfere with estimating the impact of reduced surgical radicality. Second, even though the two experimental arms were well balanced for FIGO stage, cervical diameter, histologic type, parametrial involvement, and lymph node metastases, randomization was not stratified for current validated prognostic factors (i.e., tumor size and LVSI), making it impossible to separately analyze the impact of modified RH in the subgroup of patients with low-risk disease. Third, patients were enrolled between 1987 and 1993, when MRI, PET, and SLN biopsy had not yet been introduced for parametrial and nodal evaluation. This justifies the high proportion of patients with parametrial involvement (25 vs. 27%) and positive lymph nodes (27% vs. 23%) found by the investigators. In 2014, Ditto et al. published a retrospective observational study describing the oncologic outcomes in 127 patients who underwent Piver class II RH compared to a historical cohort of 202 patients who received Piver class III RH [26]. The parametrial and lymph node involvement rates were similar in the two groups. The less radical approach provided comparable oncologic outcomes to class III RH. This is surprising if we consider that adjuvant BRT was performed in most patients in the historical cohort, regardless of pathologic factors on surgical specimens. Interestingly, a more radical procedure did not better control the local relapse rate. Overall, the recurrence rate was 12.8%, with a rate of 7.1% and 16.3% for class II and class III, respectively. In detail, pelvic relapse rates were 11.1% and 36.4% for class II and class III, respectively, whereas vaginal recurrence occurred at rates of 33.3% in class II and 21.2% in class III.

### 2.5. Current Guidelines and Risk Factors

Despite limitations in these data and the fact that differences in opinion still exist on the amount of parametrium that should be removed at surgery, surgical management of cervical carcinoma has nearly universally changed, abandoning the “one-fits-all” concept in favor of tailored surgical radicality. The recently updated 2023 ESGO/ESTRO/ESP Guidelines [2] for the management of patients with cervical cancer state that the type of radical hysterectomy (extent of parametrial resection, Q-M type A-C2) should be guided by the presence of three prognostic risk factors: tumor size, maximum stromal invasion, and LVSI. The guidelines are used to categorize patients at low, intermediate, and high risk of recurrence and treatment failure (Table 3). Interestingly, more than one option is proposed for each risk category, with classical RH (i.e., Q-M type C2) reserved as an alternative to Q-M type C1 only in patients at high risk of treatment failure.

However, it is worth noticing that, among the risk factors mentioned above, only tumor size can accurately be assessed pre-operatively by clinical evaluation, MRI, and transvaginal ultrasound (TV-US) imaging. In contrast, conclusive information about LVSI and depth of stromal invasion need histologic examination. Pre-operative histologic diagnosis of clinically evident cervical cancer is routinely performed by cervical biopsy that is inconclusive for LVSI and cannot give any information about stromal invasion depth. On the other hand, conization would provide more details on LVSI but is often omitted in pre-operative diagnostic management. However, it cannot comprehensively evaluate the depth of stromal invasion, which can be truly assessed only if a simple trachelectomy is performed with the removal of the entire width of cervical stroma.

As previously observed, we believe that depth of invasion does not give consideration to cervical size and tumor location in the cervix, and poorly represents the risk of parametrial invasion. Conversely, in agreement with Landoni et al.’s previous findings [18], we think the tumor-free distance (TFD), measured as the minimum distance of uninvolved stroma between the tumor and the peri-cervical ring, is a better predictor (Figure 1). Recently, TFD has emerged again in a retrospective cohort of 379 patients treated by primary surgery for stage IA–IIB cervical cancer, published by Cibula et al. [32]. They found a TFD ≤ 3.5 mm as an independent factor for recurrence (HR 4.58 (1.52; 13.80)). The authors even found an inverse relationship between TFD and the presence of positive lymph nodes, suggesting its implementation for recurrence risk stratification and primary treatment triage.

## 3. Low Risk Cervical Cancer: Does Tailoring Mean Abandoning Radicality?

Given the relationship between morbidity and the extent of radical hysterectomy, gynecologic oncologists have begun questioning if a subgroup of patients with low risk disease could be treated without performing parametrectomy.

### 3.1. On the Shoulders of Giants: Learning from the Past

In the first prospective study addressing this issue, Stark et al. enrolled 210 patients with stage IB cervical carcinoma between 1971 and 1979, who alternatively underwent Wertheim’s procedure or the less radical Galvin–TeLinde hysterectomy [33]. In their series, the authors reported similar recurrence (20 vs. 22%, respectively) and survival rates (5-year OS: 72 vs. 78%, respectively) between the two arms [34]. However, this study was subjected to criticism. First, randomization was not performed, and assignment to each experimental arm was performed in an alternating manner. This led to a severe imbalance in prognostic factors in the two groups, with parametria and pelvic lymph nodes less frequently involved in the less radical group. Second, many patients underwent adjuvant treatment, precluding the possibility of evaluating the real impact of reduced radicality. Adjuvant radiotherapy may have balanced the detrimental effect of less radical surgery [31].

Given these limitations, many authors started exploring less radical surgical options for early-stage cervical cancer. Multiple retrospective studies showed very low rates of parametrial invasion in patients with early-stage cervical carcinoma with favorable prognostic features [16,17,35,36,37].

In 1995, Kinney et al. evaluated 387 patients with squamous cervical carcinoma treated by radical hysterectomy. They found that none of the 83 patients with tumor size less than 2 cm and without LVSI had parametrial involvement, either via direct invasion or discontinuously in parametrial nodes [35]. Subsequently, Covens et al. [36] evaluated the incidence and risk factors for parametrial invasion in 842 patients with clinical IA2–IB1 cervical cancer who underwent either RH or radical trachelectomy. In their series, the 33 patients with parametrial involvement had older age, larger and more deeply invasive tumors, higher frequency of LVSI, and were more likely to have pelvic lymph node involvement. In patients with tumor size < 2 cm, no LVSI, depth of invasion < 10 mm, and negative pelvic nodes, the incidence of parametrial invasion was 0.6% (3/536). It is worth noting that most of the patients in this study underwent modified RH or modified radical trachelectomy and, thus, did not have their parametria entirely removed at surgery. However, 5-year RFS was 89% in patients with IB1 disease, and only 14% received adjuvant RT. The authors concluded that it was unlikely that this population’s low parametrial invasion rate could be due to false negatives. Furthermore, assuming the random distribution of nodes within the medial and lateral parametria, they estimated a theoretical incremental benefit in survival of only 0.2% if a classical overmodified RH would have been performed. Similar results were reported by Wright et al., Frumovitz et al., and, more recently, by Kodama et al. [16,17,37]. In the former study [17], authors retrospectively analyzed 594 patients with IA1–IIA cervical carcinoma treated between 1989 and 2005 with either Piver class II (3.2%) and Piver class III RH (96.8%), They found a 0.4% incidence of parametrial invasion in the subset of patients with negative lymph nodes, tumor size < 2 cm, and absence of LVSI. Similarly, in the series published by Frumovitz and colleagues [16], of the 350 patients who underwent Piver class III RH for IA2–IB1 cervical carcinoma, none of those with a tumor smaller than 2 cm, negative pelvic nodes, and no LVSI had parametrial involvement. Lastly, Kodama et al. [37] analyzed surgical specimens from 200 patients who presented IB1 cervical cancer and were treated with Piver class III RH. They aimed to determine factors predicting parametrial spread and to define a subgroup of patients at low risk for parametrial invasion. Overall, parametrial infiltration was found in 10% of their series, but decreased to 0.0% in the subset of node-negative patients younger than 50 with a tumor less than 2 cm in size, no LVSI, and depth of invasion < 10 mm.

Despite some differences in study design and results, when analyzed comprehensively, these studies demonstrate that the rate of parametrial involvement is less than 1% in patients with low-risk features. Therefore, there could be a subset of patients with early-stage cervical cancer for whom RH could be avoided. Further data from two more recent large retrospective studies strengthened these findings. Derks et al. [38] evaluated the impact of the type of surgery on recurrence and disease-free survival in 2124 patients who underwent primary surgical treatment for FIGO stage I–IIA between 1982 and 2011. In their series, only 34% of patients received adjuvant radiotherapy. Other than conventional prognostic variables, such as LVSI, tumor size, depth of invasion, and nodal involvement, the radicality of surgery was an independent prognostic variable on recurrence rate and survival. The most significant impact was observed in patients with tumor size > 4 cm. Conversely, in patients with a tumor diameter of less than 20 mm, a diminished extent of parametrectomy did not result in a worse oncological outcome. Similarly, a population-based study published by Tseng et al. [39] evaluated 2571 FIGO stage IB1 cervical cancer patients and showed no impact on survival of more radical surgery.

In 2012, Landoni et al. [40] published the first RCT prospectively evaluating the oncological impact of non-radical surgery in treating early-stage cervical carcinoma. Between 1981 and 1986, 125 patients with FIGO stage IB1 and IIA1 cervical carcinoma were randomized to a Piver class I or a Piver class III RH plus bilateral pelvic lymphadenectomy. No significant differences in disease-free (15-year DFS: 70% in class I RH vs. 86% in class III RH) and overall survival rates (15-year OS: 74% in class I RH vs. 81% in class III RH) were observed. Additionally, there was no impact of the extent of radicality on the pattern of recurrences, with 6 vs. 4 central pelvic relapses in class I and class III groups, respectively. As expected, the type of surgery influenced the morbidity rate. Notably, 84% of patients who underwent class III RH experienced at least one grade 2–3 complication, compared to 45% in the class I group. The most significant difference was observed among urologic complications, all recorded in the class III arm. However, a significant difference in OS was found in favor of more radical surgery in the subset of patients with tumor size > 3 cm, with a 15-year survival of 74% vs. 96% in the class I and class III arms, respectively. The principal limit of this study was the very high frequency of adjuvant radiotherapy administered in both arms, since the author’s attempt to evaluate the real impact of parametrectomy on oncologic outcomes probably vanished.

### 3.2. The Future Is Here

To date, it is a firm oncologic principle that the best way to treat cancer is to use a single approach to reduce complications and save other treatment options in case of failure of the first choice. Therefore, the selection of patients eligible for a less radical surgical approach is essential to avoid the need for subsequent adjuvant therapy.

Recently, two prospective studies have been published evaluating the oncological safety of non-radical surgery in selected cases of early cervical cancer. The Concerv trial [41] was a prospective single-arm multicenter observational study to assess the conization or simple hysterectomy feasibility and oncological safety in women with low-risk early-stage cervical carcinoma. In total, 100 patients with stage IA2IB1 cervical squamous carcinoma or adenocarcinoma, tumor size < 2 cm, no LVSI, depth of invasion < 10 mm, negative conization margins, and negative pre-operative imaging were enrolled. Patients with an unexpected post-operative diagnosis of invasive cancer after a simple hysterectomy were also eligible if inclusion criteria were met and the margins were negative on the hysterectomy specimen. In the 40 patients who underwent simple hysterectomy with bilateral pelvic lymphadenectomy, 3 had positive pelvic lymph nodes and subsequently received adjuvant radio-chemotherapy. In this subgroup, no patients developed recurrent disease. Conversely, of the 16 patients who had received simple hysterectomy with a post-operative diagnosis of occult invasive cervical carcinoma, 2 developed recurrence, and 1 died of the disease. Of note, none of the recurrences developed in the pelvis, suggesting that treatment failure was independent of the radicality of surgery. The LESSER trial [42] was a Brazilian single-blind randomized phase II non-inferiority trial, comparing simple versus type B2 RH plus systematic pelvic lymphadenectomy in the treatment of early-stage cervical cancer. Between 2015 and 2018, they enrolled 40 patients with FIGO 2009 stage IA2IB1 cervical squamous-, adenosquamous- or adenocarcinoma and tumor size ≤ 2 cm. The median follow-up was 52 months. The 3-year disease-free survival was 95% and 100% after simple and modified RH, respectively, with corresponding 5-year overall survival rates of 90% and 91%. However, this study presented some significant limitations. First, patients included in the study were not strictly “low risk”, since tumor characteristics such as LVSI and depth of invasion were not considered exclusion criteria. Second, pre-operative imaging was not systematically performed for local spread and distant metastasis assessment. This led to inaccurate clinical tumor size estimation, parametrial, and nodal involvement in 25%, 5%, and 7.5% of cases, respectively. Due to these limitations, it is impossible to generalize the author’s findings to current clinical practice in developed countries. Third, adjuvant therapy was administered in 25% of patients, mainly due to a loose adherence to GOG criteria in using adjuvant chemo- and/or radiotherapy. Interestingly, despite these limitations, the recurrence rate was low (1/40), highlighting the feasibility and safety of simple extra-fascial hysterectomy in their series.

The results of the Radical versus Simple Hysterectomy and Pelvic Node Dissection with Low-Risk Early-Stage Cervical Cancer (SHAPE) Trial (NCT01658930) were presented at the 2023 ASCO Annual Meeting [43], hypothetically providing practice-changing evidence for the management of early-stage low-risk cervical carcinoma. The SHAPE Trial was a large multicenter non-inferiority randomized phase III study, comparing simple vs. radical hysterectomy plus pelvic lymphadenectomy in patients with FIGO 2009 IA2–IB1 cervical carcinoma. Inclusion criteria were tumor size < 2 cm and limited stromal invasion, defined as a depth of invasion < 10 mm on LEEP/cone biopsy and/or < 50% depth involvement on MRI, irrespective of LVSI. In the simple and radical hysterectomy approaches, the 3-year pelvic recurrence rate was 2.52% vs. 2.17%, 3-year extra-pelvic recurrence-free survival 98.1% vs. 99.7% and overall survival was 99.1% vs. 99.4%. The rate of adjuvant therapy in both arms was similarly low (9.2% vs. 8.4%, respectively). As expected, the rate of bladder and ureter injuries, acute and late urinary retention, and urinary incontinence were lower with simple hysterectomy. Additionally, improved quality of life and sexual health outcomes were observed in this subset of patients.

## 4. Discussion

In the present research, we reviewed current evidence on the role of radical parametrectomy in the management of early-stage cervical carcinoma. RH has been the cornerstone of surgical treatment of cervical carcinoma in the last century. However, no agreement exists about the prevailing way of parametrial invasion, and the literature is conflicting regarding the extent of parametrectomy needed to achieve adequate surgical radicality. Although the two historical RCTs that compared less extensive with more extensive radical surgery could not assess the real impact of surgical radicality, their results guided a near-universal shift in the surgical management of early-stage cervical cancer. Current guidelines state that the extent of parametrial resection should be tailored according to pre-operative risk stratification, and classical radical hysterectomy (i.e., Q-M type C2) is reserved as an alternative to Q-M type B2-C1 only in patients at high risk of treatment failure. The risk definition reported in the recently updated 2023 ESGO/ESTRO/ESP Guidelines [2] is based on three prognostic risk factors: tumor size, maximum stromal invasion, and LVSI. However, only tumor size can accurately be assessed pre-operatively, whereas conclusive information about LVSI and depth of stromal invasion need a histologic examination of the surgical specimen. Furthermore, it is worth noting that two other critical prognostic parameters are not considered. First, depth of invasion poorly represents the risk of parametrial invasion, as it does not consider the cervical size and tumor location in the cervix. Conversely, the minimum distance of uninvolved stroma between the tumor and the peri-cervical ring (TFD) seems to be a better predictor of parametrial invasion and an independent factor for recurrence [18,32]. Second, the tumor growth pattern within the cervical stroma is not considered to triage patients and plan the extent of parametrectomy. In Landoni et al. [18] series, tumor spread within cervical stroma was found in all directions, suggesting the possibility of tailoring the removal of the most lateral part of the lateral parametria according to the tumor’s growth direction, especially when the anterior and/or posterior quadrants of the cervical stroma are involved.

The cornerstone of curative oncological surgery is complete tumor resection with clear margins, and further surgical or adjuvant treatment is warranted if adequate free margins are not achieved. Since tumor-positive margins are invariably associated with recurrence and poor prognosis, the minimal tumor-free margin is an essential clinical issue in several tumor types (i.e., melanoma, vulvar and head-and-neck carcinomas, breast cancer, soft tissue sarcomas) [44,45,46,47,48]. Conversely, the importance of surgical margins at the time of radical hysterectomy has yet to be properly defined. There is no universally accepted definition of “close” margins for cervical cancer and no consensus exists on its effects on oncologic outcomes [49]. Since the anterior parametrium is anatomically less represented than the lateral one, we believe that anterior radicality may not achieve adequate surgical margins in those patients with reduced TFD in the anterior quadrants. Although further prospective data are needed to draw conclusions on this matter, we believe these patients are at high risk of a cut-through procedure or inadequate surgical treatment and may benefit from primary concomitant chemo-radiation.

One of modern oncology’s pillars is reducing treatment-related morbidity without impairing oncologic outcomes. Many historical retrospective series have addressed whether any subgroup of patients could be treated, avoiding radical parametrectomy and its sequelae. In patients with low-risk characteristics, the parametrial involvement rate was less than 1%, suggesting that a subgroup of patients could be adequately cured, avoiding RH. The results of two prospective trials, the Concerv trial [41] and the LESSER trial [42], further supported the oncological safety of non-radical surgery in selected cases of early cervical cancer. Accordingly, the recently presented SHAPE Trial [43] seems to provide practice-changing evidence for the management of early-stage low-risk cervical carcinoma. Although we have to wait for the final manuscript, it may give a definitive answer for patients with early-stage low-risk cervical carcinoma, demonstrating that simple hysterectomy does not jeopardize oncological outcomes compared to RH, while presenting fewer complications. The GOG 278 Trial (NCT01649089) is another ongoing prospective trial evaluating the impact of non-radical surgery (simple hysterectomy or conization) in patients treated for stage IA1 LVSI+, IA2, and IB1 carcinoma of the cervix. The primary objective of this trial is to evaluate changes in functional outcomes of bowel, bladder, and sexual function before and after non-radical surgical treatment. Additionally, the site of recurrence, overall, and recurrence-free survival were included in the secondary objectives. In case the results will be in line with the SHAPE Trial, we expect the standard of care in the treatment of low-risk early-stage cervical carcinoma to change from RH to conization, simple trachelectomy, or simple hysterectomy with pelvic lymph-node status assessment.

## 5. Conclusions

Curative surgical treatment of early-stage cervical cancer does not implicate inevitably extended radical parametrectomy. In the era of personalized medicine, surgical radicality and the choice of primary treatment, whether surgical or not, should be tailored according to the patient’s tumor characteristics. Well-established risk factors have been extensively reported as predictive of treatment failure. Although little data are available addressing this issue, we believe that implementing the classical risk factors with TFD and evaluating the tumor’s growth pattern within the cervix will guide the clinician in tailoring the treatment. This would mean better triage of the patients to the adequate surgical procedure, to achieve safe clear surgical margins, and better identification of patients who should be excluded from surgery and be transitioned directly to exclusive chemo-radiation.

## Figures and Tables

**Figure 1 jpm-13-01292-f001:**
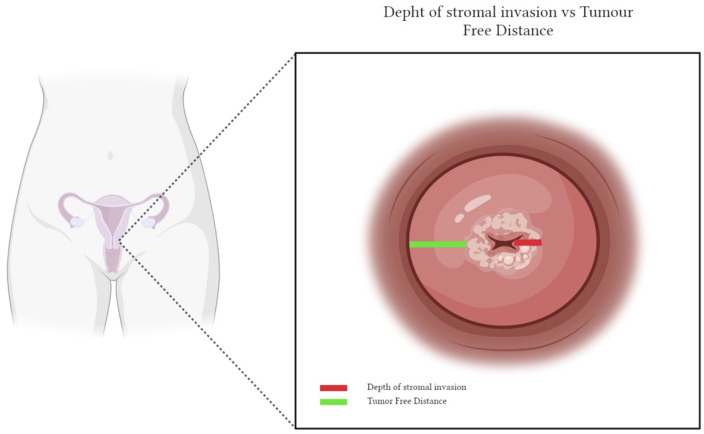
Graphical representation of depth of invasion and TFD.

**Table 1 jpm-13-01292-t001:** The 1974 Piver–Rutledge classification of radical hysterectomy [6].

Class	Parametria	Ureter	Vagina
I	Extra-fascial total hysterectomy	Ureter is identified.	No vaginal cuff
II	Modified radical hysterectomy. Uterine arteries are dissected medially from the ureter; uterosacral ligaments dissected midway by sacral insertion; cardinal ligaments dissected op to medial half.	Ureters are dissected from the lateral parametrium but not anteriorly from pubo-vesicle ligament.	Removal of upper ⅓ vagina
III	Classical radical hysterectomy. Uterine artery is ligated at its origin. Uterosacral ligaments are excised at their sacral origin. Cardinal ligaments excised as close to pelvic wall as possible.	The ureter is dissected from the pubo-vesicle ligament superiorly, inferiorly, and medially. A small lateral portion of the pubo-vesicle artery is preserved.	Removal of upper ½ of vagina
IV	Uterine arteries, cardinal, and uterosacralcardinal and uterosacral ligaments are treated as for classical RH. Umbilical vesical artery is sacrificed.	Ureter is completely dissected from pubo-vesicle ligament.	Vaginal cuff ¾ of vagina
V	As above, with addition of excision of a portion of ureter/bladder.	As above.	As above

**Table 2 jpm-13-01292-t002:** Querleu–Morrow classification of radical hysterectomy [7,8].

Type	Lateral Parametrium	Ventral Parametrium	Dorsal Parametrium
A	Halfway between the cervix and ureter.	Minimal excision	Minimal excision
B1	At the level of ureteral bed.	Partial excision of the vesicouterine ligament.	Partial resection
B2	Same as B1 plus paracervical lymphadenectomy.	Partial excision of the vesicouterine ligament.	Partial resection
C1	Transversally at the iliac vessels. Preservation of the caudal part.	Removal of the vesicouterine ligament at the bladder. Proximal part of the vesicovaginal ligament.	At the rectum
C2	At the level of the medial aspect of iliac vessels (inclusion of the caudal part).	At the bladder (bladder nerves are sacrificed).	At the sacrum (hypogastric nerve is sacrificed)
D	At the pelvic wall, together with resection of components of the pelvic sidewall and/or internal iliac vessels.	At the bladder.	At the sacrum

**Table 3 jpm-13-01292-t003:** Risk groups according to local prognostic factors and suggested type of radical hysterectomy (RH) [2].

Risk Class	Tumor Size	LVSI	Depth of Invasion	Suggested Type of RH
Low	<2 cm	−	Inner 1/3	A/B1
Intermediate	<2 cm	+	Any	B2/C1
	>2 cm	−	Any	
High	>2 cm	+	Any	C1/C2

## Data Availability

Not applicable.

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
