# Peer review of "Radical Hysterectomy in Early-Stage Cervical Cancer: Abandoning the One-Fits-All Concept"

_jpm, 2023, doi:10.3390/jpm13091292_

Round 1

Reviewer 1 Report

Dear authors, really you did a good job; your manuscript is thought-provoking. The text issue is important and will help other clinicians to attend to patients with cervical cancer. However, some imprecisions need improvements.

Detailed remarks:

You are mixing British with American English (personalization; randomized). Please, you must select one of them. Your text is so important, that needs perfect language.

The text contains some misspelled words. Look at line 290; the incorrect word is “thatthe.” Your text needs to look perfect.

In manuscripts about surgical aspects, the use of images is essential. The use of illustrations increases comprehension. Thus, it is better to add figures to Tables 1 and 2. Remember, your work is not only for experts but also for young clinicians.

Discussion is poor. You abused of literature review. You must begin with your main finding. Later, use your arguments to present the value of your work. Do not forget to explain your study weakness and future aspect to address. Remember, the discussion should not repeat the results; however, it must provide a detailed interpretation of data to explain the significance of your ideas.

Your conclusion is too long. The study’s main conclusions may be presented in a clear, short, and strong sentence.

You do not follow the journal’s instructions for authors. The journals´ names are incorrectly abbreviated in reference: 2, 6,11,12,17, 22, 25, 28, 32, 34, 36, and 40. You are using improperly the term “et al” in references: 1-4,13,14, 16, 18, 22, 26, 30, 32, 33, 35-37, 39, 41, and 42. Remember to list the first ten authors followed by et al. There are references with incomplete information: 15, 20, 24.  

Dear authors, your research is interesting and deserves to be published; however, the text has some flaws. You must hurry up to correct all your flaws.

You are mixing British with American English (personalization; randomized). Please, you must select one of them. Your text is so important, that needs perfect language.

The text contains some misspelled words. Look at line 290; the incorrect word is “thatthe.” Your text needs to look perfect.

Author Response

Dear Reviewer,

Thank you for your precise and punctual response. We tried our best to improve our article following your suggestions.

We are no longer mixing American and British English. The paper is now in American English (personalization, randomized, tumor…). We also correct the misspelled words that you found in the text.

We added Figure 1 to describe better the difference between the depth of stromal invasion and the tumor-free distance. We added a few references to guide the young colleagues in learning about the surgical anatomy of cervical cancer (see references n. 9-10-11)

We separated the discussion and the conclusion. In the first paragraph, we explain the findings of our literature review, highlight the current literature's weakness, and give suggestions for improving (TFD and evaluation of the tumor growth pattern in the cervical stroma). We also highlight in the conclusion what we consider essential for improving the management of patients with early-stage cervical cancer. In the second paragraph of the discussion, we stress the importance of free-surgical margins and hypothesize which patients should not be candidates for surgery (to avoid multiple treatments). In the third paragraph, we describe the important advances the literature makes in treating these patients, suggesting that we believe this is the right direction.

Per your suggestions, we used more precise, short, and solid sentences for the conclusions.

We corrected all the references as per the authors’ instructions.

We reviewed the English, the misspelled words, and all the wordy sentences we previously used (i.e., lines 187, 329…).

We added a few references to the manuscript.

Thank you again for your reviews. We now believe that our paper is a better insight into the current evidence about the role of radical parametrectomy and tailored surgery in early-stage cervical cancer treatment. We hope that our implementations have made the article suitable for publication.

Reviewer 2 Report

This is a clear, concise, well-organized, and well-written manuscript. The introduction and conclusions are clear. This manuscript should be published.

Author Response

Dear Reviewer,

thank you for your time in reviewing our manuscript

Reviewer 3 Report

The present manuscript aims to review current evidence on the role of radical parametrectomy in the management of early-stage cervical carcinoma. The authors lectured deeply about the subject, clearly in most of the article. I believe that after the English revisions, it can get better. My only point is that they review the tables. Table 1, for example, has no header.

The article needs minor grammatical revisions.

Author Response

Dear Reviewer,

We reviewed the English, the misspelled words, and all the wordy sentences we previously used (i.e., lines 187, 329…).

We reviewed the tables and added a header as per your suggestions.

Thank you for your time in reviewing our paper.